

**Seasonal changes in the D/H ratio of fatty acids of pelagic**
**microorganisms in the coastal North Sea**
Sandra Mariam Heinzelmann[#1], Nicole Jane Bale[1], Laura Villanueva[1], Danielle Sinke-Schoen[1],
Catharina Johanna Maria Philippart[2,3], Jaap Smede Sinninghe Damsté[1,4], Stefan Schouten[1,4],
Marcel Theunis Jan van der Meer[1]
[1] NIOZ Royal Netherlands Institute for Sea Research, Department of Marine Organic
Biogeochemistry, P.O. Box 59, 1790 AB Den Burg, The Netherlands
[2] NIOZ Royal Netherlands Institute for Sea Research, Department of Marine Ecology, P.O.
Box 59, 1790 AB Den Burg, The Netherlands
[3] Utrecht University, Faculty of Geosciences, Department of Physical Geography, Coastal
Processes, P.O. box 80.115, 3508 TC Utrecht, The Netherlands
[4] Utrecht University, Faculty of Geosciences, Department of Earth Sciences, Geochemistry,
P.O. Box 80.021, 3508 TA Utrecht, The Netherlands
[#]corresponding author: sandra.heinzelmann@nioz.nl
**To be submitted to Biogeosciences**
Running title: Shift in D/H ratio of fatty acids during seasonal changes in pelagic microbial
communities
Keywords: metabolism, fatty acids, photoautotrophs, heterotrophs, algal bloom, 16S rRNA
gene amplicon pyrosequencing, bacterial diversity, coastal environment, hydrogen isotopes



**Abstract**
Culture studies of microorganisms have shown that the hydrogen isotopic composition of fatty
acids depends on their metabolism, but there are only few environmental studies available to
confirm this observation. Here we studied the seasonal variability of the deuterium/hydrogen
(D/H) ratio of fatty acids in the coastal Dutch North Sea and compared this with the diversity
of the phyto- and bacterioplankton. Over the year, the stable hydrogen isotopic fractionation
factor ε between fatty acids and water ranged between -172 ‰ and -237 ‰, the algal-derived
polyunsaturated fatty acid $n$C20:5 being the most D-depleted and $n$C18:0 the least D-depleted
fatty acid. The D-depleted $n$C20:5 is in agreement with culture studies, which indicates that
photoautotrophic microorganisms produce fatty acids which are significantly depleted in D
relative to water. The $\varepsilon_{lipid/water}$ of all fatty acids showed a transient shift towards increased
fractionation during the spring phytoplankton bloom, indicated by increasing chlorophyll $a$
concentrations and relative abundance of the $n$C20:5 PUFA, suggesting increased contributions
of photoautotrophy. Time periods with decreased fractionation (less negative $\varepsilon_{lipid/water}$ values)
can be explained by an increased contribution by heterotrophy to the fatty acid pool. Our results
show that the hydrogen isotopic composition of fatty acids is a useful tool to assess the
community metabolism of coastal plankton.
**1. Introduction**
The hydrogen isotopic composition of fatty acids of microorganisms has been shown to depend
on different factors like metabolism, salinity, biosynthetic pathways, growth phase and
temperature (Dirghangi and Pagani, 2013; Fang et al., 2014; Heinzelmann et al., 2015a;
Heinzelmann et al., 2015b; Zhang et al., 2009a; Zhang et al., 2009b). While most of these factors
lead to relatively small variations in the deuterium to hydrogen (D/H) ratio of fatty acids (10-
20 ‰), differences in the central metabolism of microorganisms have a much more pronounced
effect. Both photo- and chemoautotrophs produce fatty acids depleted in D compared to growth



water with the stable hydrogen isotopic fractionation factor ε between fatty acids and water
($\varepsilon_{lipid/water}$) ranging between -150 ‰ to -250 ‰ and -250 ‰ and -400 ‰, respectively (Campbell
et al., 2009; Chikaraishi et al., 2004; Heinzelmann et al., 2015a; Heinzelmann et al., 2015b;
Sessions et al., 2002; Valentine et al., 2004; Zhang et al., 2009a; Zhang and Sachs, 2007). In
contrast, heterotrophs produce fatty acids with either a relatively minor depletion or an
enrichment in D compared to the growth water with $\varepsilon_{lipid/water}$ values ranging between -150 ‰
and +200 ‰ (Dirghangi and Pagani, 2013; Fang et al., 2014; Heinzelmann et al., 2015a;
Heinzelmann et al., 2015b; Sessions et al., 2002; Zhang et al., 2009a). The differences in
hydrogen isotopic composition of fatty acids produced by organisms expressing different core
metabolisms have mainly been attributed to the D/H ratio of nicotinamide adenine dinucleotide
phosphate (NADPH) (Zhang et al., 2009a). NADPH can be generated by a variety of different
reactions in different metabolic pathways (each associated with different hydrogen isotopic
fractionations) and is subsequently used as the main source of hydrogen in lipid biosynthesis
(Robins et al., 2003; Saito et al., 1980; Schmidt et al., 2003).
Although the metabolism of a microorganism in pure culture is reflected by the D/H ratio of its
fatty acids, it is not clear if the D/H ratio of fatty acids from environmental microbial
communities can be used to assess the 'integrated' core metabolisms in nature. Culture
conditions rarely represent environmental conditions since cultures are typically axenic and use
a single substrate, they do not take into account microbial interactions, and they test a limited
number of potential substrates, energy sources and core metabolisms. Previous studies observed
a wide range in the D/H ratio of lipids in both water column and sediment (Jones et al., 2008;
Li et al., 2009), suggesting inputs of organisms with a variety of metabolisms. So far, one
environmental study has been performed that links the D/H ratio of fatty acids from naturally
occurring microbial communities to metabolisms possibly expressed by the members of those
communities (Osburn et al., 2011). This study showed that different microbial communities



from various hot springs in Yellowstone National Park produce fatty acids with hydrogen
isotopic compositions in line with the metabolism expressed by the source organism. The D/H
ratio of specific fatty acids, which could be attributed to microorganisms expressing a specific
core metabolism, was within the range expected for that metabolism. On the other hand, the
D/H ratio of common or general fatty acids (e.g. $n$C16:0) allowed for assessing the metabolism
of the main contributors of these more general fatty acid, but not necessarily the metabolism of
the dominant community members (Osburn et al., 2011). These results show the applicability
of this new method, but the ecosystems in which it was tested (hot spring microbial
communities) are considered to be of relatively low diversity. Therefore, this method needs to
be applied and evaluated in more complex and diverse microbial communities.
Here, we studied the seasonal variability of the hydrogen isotopic composition of fatty acids
from coastal North Sea water sampled from the jetty at the Royal Netherlands Institute for Sea
Research (NIOZ) in order to examine the relationship between hydrogen isotope fractionation
in fatty acids and the general metabolism of the community. Time series studies have been
previously performed at the NIOZ jetty to determine phytoplankton and prokaryotic abundances
and composition (Alderkamp et al., 2006; Brandsma et al., 2012; Brussaard et al., 1996;
Philippart et al., 2010; Philippart et al., 2000; Pitcher et al., 2011; Sintes et al., 2013), lipid
composition (Brandsma et al., 2012; Pitcher et al., 2011), and chlorophyll $a$ concentration
(Philippart et al., 2010). Typically, the spring bloom in the coastal North Sea is predominantly
formed by *Phaeocystis globosa*, followed directly by a bloom of various diatom species, a
second moderate diatom bloom of *Thalassiosira spp.* and *Chaetoceros socialis* that occurs in
early summer and an autumn bloom is formed by *Thalassiosira spp.*, *C. socialis*, cryptophytes
and cyanobacteria (Brandsma et al., 2012; Cadée and Hegeman, 2002), although the autumn
bloom seems to have weakened over the last years (Philippart et al., 2010). The abundance of
bacteria co-varies with algal blooms and the bacteria are dominated by heterotrophs, e.g.



bacteria belonging to the *Bacteroidetes* (Alderkamp et al., 2006), using released organic matter
from declining phytoplankton blooms as carbon, nitrogen and phosphate sources. The intact
polar lipid (IPL) composition of the microbial community was shown to be composed mainly
of phospholipids, sulfoquinovosyldiacylglycerol and betaine lipids with a limited taxonomic
potential (Brandsma et al., 2012). The main source of those lipids was assumed to be the
eukaryotic plankton.
This well studied site should allow us to trace the shift from an environment dominated by
photoautotrophs during major phytoplankton blooms, towards an environment with a higher
abundance of heterotrophic bacteria following the end of the bloom. These shifts in the
community structure should be reflected in the D/H ratio of fatty acids. We, therefore, analysed
the D/H ratio of polar lipid derived fatty acids (PLFA) over a seasonal cycle and compared this
with phytoplankton composition data and abundance and information on the bacterial diversity
obtained by 16S rRNA gene amplicon sequencing.
**2. Material and Methods**
**2.1. Study site and sampling**
Surface water samples were collected from September 2010 until December 2011 from the
NIOZ sampling jetty in the Marsdiep at the western entrance of the North Sea into the Wadden
Sea at the island of Texel (53°00'06" N 4°47'21" E). Samples were taken during high tide to
ensure that the water sampled was North Sea water.
For lipid analysis measured volumes of water (ca. 9-11 L) were filtered consecutively, without
pre filtration, through pre-ashed 3 and 0.7 µm pore size glass fibre filters (GF/F, Whatman; 142
mm diameter) and stored at -20 ºC until lipid extraction. For DNA analysis approximately 1 L



seawater was filtered through a polycarbonate filter (0.2 µm pore size; 142 mm diameter;
Millipore filters) and stored at -80 ℃ until extraction.
Salinity measurements were done during the time of sampling with either an Aanderaa
Conductivity/Temperature sensor 3211 connected to an Aanderaa datalogger DL3634
(Aanderaa Data Instruments AS, Norway) or a Refractometer/Salinometer Endeco type 102
handheld (Endeco, USA).
For chlorophyll *a* measurements 500 mL sea water was filtered through a 47 mm GF/F filter
(0.7 µm pore size, Whatman, GE Healthcare Life Sciences, Little Chalfont, UK) and
immediately frozen in liquid nitrogen. Samples were thawed and homogenised with glass beads
and extracted with methanol. Chlorophyll *a* concentration was measured with a Dionex high-
performance liquid chromatography (HPLC) (Philippart et al., 2010).
Water samples for salinity versus $\delta D_{water}$ calibration (see below) were sampled weekly between
March and September 2013 at high tide. Salinity was determined using a conductivity meter
(VWR EC300) calibrated to IAPSO standard seawater of salinities 10, 30, 35 and 37.

**2.2. Polar lipid derived fatty acids**

Filters were extracted for IPLs and eventually fatty acid analysis. The 0.7 µm filters did not
yield enough total lipid extract for analysis. Therefore, only fatty acids obtained from the 3 µm
filters were analysed. Due to fast clogging of the filters and a corresponding decrease of the
pore size (Sørensen et al., 2013), the 3 µm filters will most likely contain most of the
microorganisms present in North Sea water. Freeze dried filters were extracted via a modified
Bligh-Dyer method (Bligh and Dyer, 1959; Rütters et al., 2002) with methanol
(MeOH)/dichloromethane (DCM)/phosphate buffer (2:1:0.8, vol/vol/vol) using ultrasonication
(Heinzelmann et al., 2014). Approximately 0.5 - 1 mg of the Bligh-Dyer extract (BDE) was
separated into a neutral and polar lipid fraction using silica column chromatography





(Heinzelmann et al., 2014). The BDE was added onto a DCM pre-rinsed silica column (0.5 g;
activated for 3 h at 150°C) and eluted with 7 mL of DCM and 15 mL of MeOH. The resulting
fractions were dried under nitrogen and stored at -20 °C. PLFAs were obtained via
saponification of the MeOH fraction with 1 N KOH in MeOH (96%). The samples were
refluxed at 140 °C for 1 h. Afterwards the pH was adjusted to 5 with 2 N HCl/MeOH (1/1),
bidistilled $H_2O$ and DCM were added. The $MeOH/H_2O$ layer was washed twice with DCM, the
DCM layers were combined and dried over $Na_2SO_4$. The sample was dried under nitrogen and
stored in the fridge. The PLFAs were methylated with boron trifluoride-methanol ($BF_3$-MeOH)
for 5 min at 60 °C. Afterwards $H_2O$ and DCM were added. The $H_2O/MeOH$ layer was washed
three times with DCM, and potential traces of water were removed over a small $Na_2SO_4$ column
after which the DCM was evaporated under a stream of nitrogen. In order to obtain a clean
PLFA fraction for isotope analysis, the methylated extract was separated over an aluminium
oxide ($Al_2O_3$) column, eluting the methylated PLFAs with three column volumes of DCM. For
identification of the position of double bonds in unsaturated fatty acids, the methylated PLFAs
were derivatised with dimethyldisulfide (DMDS) (Nichols et al., 1986). Hexane, DMDS and
$I_2$/ether (60 mg/mL) were added to the fatty acids and incubated at 40 °C overnight. After adding
hexane, the iodine was deactivated by addition of a 5% aqueous solution of $Na_2S_2O_3$. The
aqueous phase was washed twice with hexane. The combined hexane layers were cleaned over
$Na_2SO_4$ and dried under a stream of nitrogen. The dried extracts were stored at 4 °C.
**2.3. Fatty acid and hydrogen isotope analysis**
The fatty acid fractions were analysed by gas chromatography (GC) using an Agilent 6890 gas
chromatograph with a flame ionization detector (FID) using a fused silica capillary column (25
m x 320 µm) coated with CP Sil-5 (film thickness 0.12 µm) with helium as carrier gas. The
temperature program was as follows: initial temperature 70 °C, increase of temperature to 130
°C with 20 °C min[-1], and then to 320 °C with 4 °C min[-1] which was kept for 10 min. Individual



compounds were identified using GC/mass spectrometry (GC/MS) and the position of the
double bonds in unsaturated fatty acids was determined after derivatisation with
dimethyldisulfide (Heinzelmann et al., 2015b).
Hydrogen isotope analysis of the fatty acid fraction was performed by GC thermal conversion
isotope ratio monitoring MS (GC/TC/irMS) using an Agilent 7890 GC connected via Thermo
GC Isolink and Conflo IV interfaces to a Thermo Delta V MS according to Chivall et al. (2014).
Samples were injected onto an Agilent CP-Sil 5 CB column (25 m × 0.32 mm ID; 0.4 µm film
thickness; He carrier gas, 1.0 mL min$^{-1}$). The GC temperature program was 70 °C to 145 °C at
20 °C min$^{-1}$, then to 320 °C at 4 °C min$^{-1}$ where it was kept for 15 min. Eluting compounds
were converted to $H_2$ at 1420°C in an $Al_2O_3$ tube before introduction into the mass spectrometer.
The $H^{3+}$ correction factor was determined daily and was constant at 5.3±0.2. A set of standard
*n*-alkanes with known isotopic composition (Mixture B prepared by Arndt Schimmelmann,
University of Indiana) was analyzed daily prior to analyzing samples in order to monitor the
system performance. Samples were only analyzed when the n-alkanes in Mix B had an average
deviation from their off-line determined value of <5 ‰. An internal standard, squalane (δD = -
170 ‰) was co-injected with each fatty acid sample fraction in order to monitor the precision
of the measurements over time with δD = -164±4 ‰. The δD of the individual fatty acids was
measured in duplicates and corrected for the added methyl group (Heinzelmann et al., 2015b).
δD of water samples was determined by elemental analysis/TC/irMS (EA/TC/irMS) according
to Chivall et al. (2014).
**2.4. Phytoplankton abundance and diversity**
Phytoplankton samples were preserved with acid Lugol's iodine, and cells were counted with a
Zeiss inverted microscope using 3 mL counting chambers. Most algae were identified to species
level, but some were clustered into taxonomic and size groups (Philippart et al., 2000). For each



sampling date in the period from September 2010 to December 2011, the densities of the most
abundant phytoplankton species or species' groups were calculated. The three most dominant
algal species (or groups) together comprised, on average, more than 60% of the total numbers
of marine algae in the Marsdiep during this study period.

**2.5. DNA extraction**

The 0.2 µm polycarbonate filters were defrosted and cut into small pieces with sterile scissors
and then transferred into a 50 mL falcon tube. Filter pieces were lysed by bead-beating with ~1
g of sterile 0.1 mm zirconium beads (Biospec, Bartlesville, OK) in 10 mL RLT buffer (Qiagen)
and 100 µL β-mercaptoethanol for 10 min. 1/60 volume RNase A (5 µg/µL) was added to the
lysate, incubated for 30 min at 37 °C and afterwards cooled down for 5 min on ice. The lysate
was purified with the DNeasy Blood and Tissue kit (Qiagen, Hilden). DNA was eluted with 3x
100 µL AE buffer, the eluates pooled and reconcentrated. DNA quality and concentration was
estimated by Nanodrop (Thermo Scientific, Waltham, MA) quantification.

**2.6. 16S rRNA gene amplicon sequencing and analysis**

The general bacterial diversity was assessed by 16S rRNA gene amplicon pyrotag sequencing.
The extracted DNA was quantified fluorometrically with Quant-iT$^{TM}$ PicoGreen® dsDNA
Assay Kit (Life Technologies, The Netherlands).
PCR reactions were performed with the universal (Bacteria and Archaea) primers S-D-Arch
0519-a-S-15 (5'-CAG CMG CCG CGG TAA-3') and S-D-Bact-785-a-A-21 (5'-GAC TAC
HVG GGT ATC TAA TCC-3') (Klindworth et al., 2012) adapted for pyrosequencing by the
addition of sequencing adapters and multiplex identifier (MID) sequences. To minimize bias
three independent PCR reactions were performed containing: 16.3 µL $H_2O$, 6 µL HF Phusion
buffer, 2.4 µL dNTP (25 mM), 1.5 µL forward and reverse primer (10 µM; each containing an



unique MID tail), 0.5 µL Phusion Taq and 2 µL DNA (6 ng/µL). The PCR conditions were
following: 98 ℃, 30 s; 25× [98 ℃, 10 s; 53 ℃, 20 s; 72 ℃, 30 s]; 72 ℃, 7 min and 4 ℃, 5 min.
The PCR products were loaded on a 1% agarose gel and stained with SYBR® Safe (Life
Technologies, The Netherlands). Bands were excised with a sterile scalpel and purified with
Qiaquick Gel Extraction Kit (QIAGEN, Valencia, CA) following the manufacturer's
instructions. PCR purified products were quantified with Quant-iT$^{TM}$ PicoGreen® dsDNA
Assay Kit (Life Technologies, The Netherlands). Equimolar concentrations of the barcoded
PCR products were pooled and sequenced on GS FLX Titanium platform (454 Life Sciences)
by Macrogen Inc. Korea.
Samples were analyzed using the QIIME pipeline (Caporaso et al., 2010). Raw sequences were
demultiplexed and then quality-filtered with a minimum quality score of 25, length between
250−350 bp, and allowing maximum two errors in the barcode sequence. Sequences were then
clustered into operational taxonomic units (OTUs, 97% similarity) with UCLUST (Edgar,
2010). Reads were aligned to the Greengenes Core reference alignment (DeSantis et al., 2006)
using the PyNAST_algorithm (Caporaso et al., 2010). Taxonomy was assigned based on the
Greengenes taxonomy and a Greengenes reference database (version 12_10) (McDonald et al.,
2012; Werner et al., 2012). Representative OTU sequences assigned to the specific taxonomic
groups were extracted through classify.seqs and get.lineage in Mothur (Schloss et al., 2009) by
using the Greengenes reference and taxonomy files. The 16S rRNA gene amplicon reads (raw
data) have been deposited in the NCBI Sequence Read Archive (SRA) under BioProject number
PRJNA293285.
**2.7. Phylogenetic analyses**
The phylogenetic affiliation of the 16S rRNA gene sequences was compared to release 119 of
the Silva NR SSU Ref database (http://www. arb-silva.de/; Quast (2012)) using the ARB





software package (Ludwig et al., 2004). Sequences were added to the reference tree supplied
by the Silva database using the ARB Parsimony tool.
**3. Results**
Suspended particulate matter (SPM) of North Sea coastal water was obtained during a period
from August 2010 - December 2011, covering a complete annual cycle, in approximately
biweekly resolution.
**3.1. Chlorophyll *a* concentration and phytoplankton abundance and diversity**
Chlorophyll *a* concentrations ranged between 0.4 and 22.2 µg L$^{-1}$ (Fig. 1; Table S1). During
late autumn, winter and early spring concentrations were low at ~4 µg L$^{-1}$. A peak in the
chlorophyll *a* concentration occurred in the beginning of April and values stayed relatively high
during this month, indicative of the spring bloom. Subsequently, the chlorophyll *a* concentration
decreased again, reaching pre-bloom levels and stayed relatively constant thereafter.
Phytoplankton diversity and abundance was determined using light microscopy and the two to
three most abundant phytoplankton species were identified and counted (Table S2). The
majority of the phytoplankton was composed of *Phaeocystis globosa*, diatoms and
cyanobacteria (Fig. 2), with the spring bloom primarily being made up of *P. globosa*. The
highest abundance of diatoms was also during spring, while the cyanobacteria reached the
highest abundance in the beginning of the sampling period from autumn until late winter and
again during summer.
**3.2. Microbial diversity**
To assess bacterial diversity, 16S rRNA gene amplicon sequencing was performed on
approximately half of the SPM samples (Table S3).



The bacteria detected consisted mainly of members of *Actinobacteria*, *Bacteriodetes*,
*Planctomycetes*, *α-Proteobacteria*, *β-Proteobacteria*, *γ-Proteobacteria* and *Verrucomicrobia*
(Fig. 3; Table S3). The majority of the reads belonged to the orders of the *Flavobacteriales*,
*Rhodobacteriales*, *Rickettsiales*, *Alteromonadales* and *Oceanospirillales*. The *Flavobacteriales*
contributed between 12 to 32 % to the total bacterial reads with a relatively constant percentage
of ~ 15 % during autumn and winter. The percentage of reads increased during early spring
with the highest values from beginning of April until the end of May. The percentage of reads
attributed to the *Flavobacteriales* decreased during summer and early autumn. Sequence reads
affiliated to the *Rhodobacteriales* (6 to 12 %) and *Rickettsiales* (3 to 17 %) were the most
represented within the *α-Proteobacteria.* The percentage of *Rhodobacteriales* reads was fairly
constant with no obvious seasonal pattern. In contrast, the percentage of *Rickettsiales* reads
followed a distinct seasonal pattern with a maximum in April (up to 17 %) and a minimum in
June (3 %). *Alteromonadales* reads made up between 9 and 17 % of all bacteria reads and were
fairly constant over the season. The percentage of *Oceanospirillales* reads were between 3 and
12 % of the total bacteria reads and show a clear maximum during mid-April (Fig. 3; Table S3).
For a more accurate taxonomic classification of the bacterial groups, sequence reads of the
*Bacteriodetes*, *α-Proteobacteria* and *γ-Proteobacteria* were extracted from the dataset and a
phylogenetic tree was constructed (Fig. S1-S3). Within the *Flavobacteriales* (*Bacteroidetes*)
the majority of the reads fell either within the *Cryomorphaceae* or the *Flavobacteriaceae* with
sequences clustering within *Fluviicola* and *Crocinitomix*, *Flavobacterium* and *Tenacibaculum*,
respectively. Within the *Rhodobacterales* (*α-Proteobacteria*) most of the reads belonged to
*Rhodobacteraceae* and sequences within this family were closely related to the genus
*Octadecabacter*. Within the *Rickettsiales* most of the reads were affiliated to the
*Pelagibacteraceae* (SAR11 cluster). The majority of the *γ-Proteobacteria* reads were classified
within the *Alteromonadales* and *Oceanospirillales*. The *Alteromonadales* reads and sequences



fell within the uncultured *HTCC2188*-isolate and *OM60*-clade and various members of the
*Alteromonadaceae*-family. The *Halomonadaceae* family comprised most of the
*Oceanospirillales* reads and additionally sequences clustered with various members of the
*Oceanospirillaceae*.

**3.3. Fatty acid distribution in North Sea SPM**

Polar lipid derived fatty acids were comprised of *n*C14:0, *n*C16:1ω7, *n*C16:0, *n*C18:0, the
polyunsaturated fatty acid (PUFA) *n*C20:5, and various unsaturated *n*C18 fatty acids (Fig. 4;
Table S4). The *n*C14:0 fatty acid followed a seasonal cycle with the lowest relative abundance
during winter, and the highest from June to August (Fig. 4a). The *n*C16:0 fatty acid was the
dominant fatty acid (21-38 %) with no clear seasonal pattern (Fig. 4c). The *n*C16:1 fatty acid
was the next most abundant fatty acid (13–35 %) with a maximum from March to April (Fig.
4b). Various unsaturated *n*C18:x fatty acids were observed throughout the season. Due to low
abundance of the individual fatty acids and co-elutions the double bond positions could not be
determined. These unsaturated fatty acids made up 9–30 % of all fatty acids (Fig. 4d). The
*n*C18:0 fatty acid had relative abundances varying between 2–18 % with the highest relative
abundance during autumn months (10–18 %) and the lowest during spring, 2–6 % (Fig. 4e). A
*n*C20:5 PUFA (Fig. 4f) was observed in most samples with the highest relative abundance
during March and April (11–14 %) and early August (18 %). Trace amounts of *n*C15:0, *i*C15:0
and *ai*C15:0 fatty acids were also detected.

**3.4. Hydrogen isotopic composition of fatty acids**

δD values of *n*C14:0, *n*C16:1ω7, *n*C16:0, *n*C18:0 fatty acids and *n*C20:5 were obtained for most
of the samples (Table S5). The D/H ratio of the other fatty acids could not be determined with
sufficient accuracy due to either incomplete separation or low abundance.



In general, $n$C14:0 and $n$C20:5 were the most depleted fatty acids with δD values ranging
between -198 to -241 ‰ and -180 to -241 ‰, respectively. The $n$C18:0 was typically the fatty
acid with the highest δD values ranging between -175 to -212 ‰ (Table S5).
**4. Discussion**
**4.1. Hydrogen isotopic fractionation expressed in fatty acids**
For the proper assessment of the impact of metabolism on the hydrogen isotopic composition
of fatty acids the hydrogen isotopic fractionation of the fatty acids versus water is required
($\varepsilon_{\text{lipid/water}}$). For this, the δD of the water ($\delta D_{\text{water}}$) at the time of sampling is needed. However,
at the time of sampling of the SPM unfortunately no water samples were taken and preserved
for δD analysis. Therefore, we used an alternative approach to estimate $\delta D_{\text{water}}$ using the salinity
of the water measured at the time of sampling. A strong correlation between salinity and $\delta D_{\text{water}}$
is generally observed in marine environments since both parameters depend on evaporation,
precipitation and freshwater influx (Craig and Gordon, 1965; Mook, 2001). To establish a local
salinity - $\delta D_{\text{water}}$ correlation, water samples were collected weekly during high tide (March to
September 2013) and salinity and $\delta D_{\text{water}}$ were measured. Indeed, a strong correlation between
salinity and $\delta D_{\text{water}}$ is observed ($R^2$=0.68; Fig. S4). Using this correlation and the salinities
measured, we reconstructed $\delta D_{\text{water}}$ values at the time of sampling of the biomass (Table 1). The
error in the estimate of $\varepsilon_{\text{lipid/water}}$ resulting from this approach is approximately 1.5 ‰, which is
less than the error in the determination of δD of the fatty acids (1-12 ‰).
All fatty acids were depleted in D compared to water with the fractionation factor $\varepsilon_{\text{lipid/water}}$
ranging from -173 to -237 ‰, all following a similar seasonal trend with the highest degree of
fractionation during spring to early summer, and early autumn (Fig. 5; Table 1). The lowest



degree of fractionation (most positive $\varepsilon_{lipid/water}$ values) was in general during late autumn and
the winter months.

**4.2. Source affects the hydrogen isotopic composition of individual fatty acids**

The $n$C20:5 PUFA is the most specific fatty acid detected in North Sea SPM and is exclusively
produced by algae (Carrie et al., 1998). The $n$C20:5 PUFA is one of the most D-depleted fatty
acids (Fig. 5), which is in agreement with culture studies that show that photoautotrophic
microorganisms produce fatty acids that are depleted in D with $\varepsilon_{lipid/water}$ values between -162
and -215 ‰, while heterotrophic microorganisms on the other hand produce fatty acids with
$\varepsilon_{lipid/water}$ values ranging between -150 to +200 ‰. Furthermore, its concentration increased at
the time of the phytoplankton bloom (Fig. 4). Interestingly, after the phytoplankton bloom,
when the abundance of pelagic algae had decreased (Fig. 4), it became more enriched in D (Fig.
5). This enrichment might be due to changes in the relative contribution of source organisms.
In diatoms $n$C20:5 PUFA can be one of the most abundant fatty acids, while *Phaeocystis*
produces it in minor amounts only (Table S6). During the spring bloom both organisms will
contribute to the fatty acid pool, while afterwards diatoms are the main source (Fig. 2; Table
S2). Another possible reason could be that after the bloom and due to nutrient limitation,
phytoplankton might use more storage products leading to an increased production of NADPH
via other pathways than photosynthesis. The NADPH produced by photoautotrophs via
photosystem I is depleted in D (Zhang et al., 2009a), while NADPH produced via the pentose
phosphate (OPP) pathway and the tricarboxylic acid (TCA) cycle is relatively enriched in D
(Heinzelmann et al., 2015b; Zhang et al., 2009a). The utilization of storage products would lead
to an increased production of NADPH via both the OPP pathway and the TCA cycle leading to
more positive $\varepsilon_{lipid/water}$ values of the $n$C20:5 PUFA after the bloom.



Of all other fatty acids $n$C14:0 was generally the most D-depleted fatty acid, possibly suggesting
a higher contribution of photoautotrophic organisms to this fatty acid. The quite similar $\varepsilon_{lipid/water}$
values of $n$C16:1 (-179 to -224 ‰) and $n$C16:0 (-178 to -215 ‰) relatively to each other suggest
similar sources for the two fatty acids. The least negative $\varepsilon_{lipid/water}$ values for $n$C18:0 suggest
that the sources of this fatty acid might differ from the other fatty acids i.e. with a higher
contribution of heterotrophs compared to the other fatty acids.
Fatty acids profiles of representatives of most members of the phytoplankton and bacterial
community observed at our site have been previously reported (Table S6) and can be used to
assess the main sources of the different fatty acid pools. The main bacterial contributors to the
$n$C16:0 and $n$C16:1ω7 fatty acids are most likely members of the *Alteromonadales* and the
*Halomonadaceae*, while the majority of bacterial contributors to the $n$C14:0 and $n$C18:0 fatty
acid are derived from the *Puniceicoccales* (Table S6). Both the *Flavobacteriales* and the
*Rhodobacteriaceae,* which make up a large part of the total bacteria reads, will hardly contribute
to the measured isotopic signal as they have been reported to produce only traces of $n$C14:0,
$n$C16:0, $n$C16:1ω7 or $n$C18:0 fatty acids (Table S6). The observed phytoplankton species are
main contributors to the $n$C14:0, $n$C16:0 and $n$C16:1ω7 fatty acid pools, but contribute
relatively little to the $n$C18:0 fatty acid pools. *Phaeocystis* produces mainly the $n$C14:0 and
$n$C16:0 fatty acids (Hamm and Rousseau, 2003; Nichols et al., 1991).
Overall, the majority of the $n$C14:0 fatty acid pool will likely be predominately derived from
photoautotrophs (Table S6), which potentially explains why the $n$C14:0 is almost always the
most depleted fatty acid. The $n$C18:0 fatty acid on the other hand, will be mainly derived from
heterotrophic bacteria (Table S6) resulting in more D enriched signal compared to that of the
$n$C14:0 fatty acid.





Culture studies have shown that chemoautotrophs produce fatty acids, which are even more
depleted in D compared to photoautotrophs. However, none of the fatty acids measured in the
North Sea SPM have $\varepsilon_{lipid/water}$ values which fall in the range of those predicted for
chemoautotrophs (-264 to -345 ‰; Heinzelmann et al., 2015b). This fits with the observation
that sequence reads of chemoautotrophic bacteria accounted for < 3 % of the total bacterial
reads (Fig. 3; Table S3), and thus it is unlikely that this metabolism plays an important role in
this environment.
**4.3. Linking seasonal changes of hydrogen isotope fractionation to changes in community**
**metabolism**
All the fatty acids showed a similar seasonal trend with the most negative ε values in spring and
the most positive ε values in the winter (Fig. 5). In order to assess the dominant metabolism of
the whole microbial community we calculated a weighted average ε of all measured fatty acids
apart from the specific $n$C20:5 PUFA. The weighted average $\varepsilon_{lipid/water}$ ($\varepsilon_{\Sigma FA}$) followed the same
seasonal trend as the $\varepsilon_{lipid/water}$ values of the individual fatty acids (Fig. 1+5), and ranged between
-180 and -225 ‰ with an average of -199 ‰.
Compared to the chlorophyll $a$ concentration, the $\varepsilon_{\Sigma FA}$ followed an opposite seasonal trend i.e.
when the chlorophyll $a$ concentration increased in early April, $\varepsilon_{\Sigma FA}$ decreased (Fig. 5). The
chlorophyll $a$ maximum in April-May indicates a spring bloom (Fig. 2), which is known to
occur annually in North Sea coastal waters (Brandsma et al., 2012; Philippart et al., 2010) and
corresponds with a shift towards more negative values for $\varepsilon_{\Sigma FA}$, as well as a high abundance of
the algal-derived $n$C20:5 PUFA (Fig. 4). It is likely that at least during the spring bloom the
majority of the fatty acids are derived from the dominant algae, i.e. *Phaeocystis* and diatoms,
which make up the majority of the bloom, leading to a D depleted signal. Thus, the observation
that the value of $\varepsilon_{\Sigma FA}$ was more negative during the spring bloom when the environment is



dominated by photoautotrophic microorganisms (Fig. 3) fits with an increased contribution by
photoautotrophs relative to heterotrophic microorganisms to the fatty acid pool. At the end of
the bloom more positive $\varepsilon_{\Sigma FA}$ values were observed, which is in agreement with an increased
abundance of heterotrophic bacterioplankton in previous studies (Sintes et al., 2013), living on
released organic material (Alderkamp et al., 2006). Interestingly, analysis of suspended
particulate matter from the California borderland basins also showed that typical bacterial fatty
acids were generally enriched in D while algal fatty acids were more depleted in D (Jones et al.,
2008), similar to what we observed here.
Thus, $\varepsilon_{\Sigma FA}$ values reflect a mixed signal derived from mainly photoautotrophic and, to a lesser
extent, heterotrophic microorganisms. Nevertheless, $\varepsilon_{lipid/water}$ values for all fatty acids remain
in the range of photoautotrophic metabolism (Heinzelmann et al., 2015b), indicating that,
overall, the fatty acids in this coastal seawater are mostly derived from phototrophic organisms.
This is in accordance with the assumption that IPLs (containing fatty acids) in coastal North
Sea waters over the annual cycle were predominantly derived from phytoplankton (Brandsma
et al., 2012). Our results show that it is possible to study whole community core metabolism in
a natural environment by determining the weighted average D/H ratio of all fatty acids.
**5. Conclusion**
A seasonal study of fatty acids derived from the coastal Dutch North Sea shows that all fatty
acids are depleted in D with δD ranging between -174 and -241 ‰. The most negative values
were observed during the spring bloom, when the biomass is dominated by photoautotrophic
microorganisms. The subsequent higher relative contribution of heterotrophs to the general fatty
acid pools leads to shift in $\varepsilon_{lipid/water}$ towards more positive values by up to 20 ‰. This shift
towards more positive values is in agreement with observations from culture studies where
heterotrophic organisms fractionate much less or even opposite to photoautotrophic organisms.



This study confirms that hydrogen isotopic fractionation as observed in general fatty acids can
be used to study the core metabolism of complex environments and to track seasonal changes
therein.
**Data availability**
Data is available on Pangea under doi:10.1594/PANGAEA.859031
**Author Contribution**
N.J. Bale helped by providing samples and helped with sampling; L. Villanueva helped with
carrying out sequencing experiments and analysis of subsequent data; D. Sinke-Schoen helped
with measuring the hydrogen isotopic composition of North Sea water samples; C. J. M.
Philippart provided chlorophyll *a* and phytoplankton data; J. S. Sinninghe Damsté, S. Schouten
and M. T. J. van der Meer helped design experiments and contributed to the manuscript as
supervisors of S.M. Heinzelmann; S.M. Heinzelmann prepared the manuscript with
contributions of all co-authors.

**Acknowledgment**
The authors would like to thank Y. A. Lipsewers, E. Svensson and K. K. Sliwinska for their
help with sampling. We would like to thank E. Wagemaakers for providing salinity data, M.
Veenstra and A. van den Oever for assistance with the phytoplankton sampling and analyses,
and M. Verweij for assistance with the GC-MS measurements. MvdM was funded by the Dutch
Organisation for Scientific Research (NWO) through a VIDI grant.



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

**Figure and tables legends**
**Figure 1**
$\varepsilon_{average}$ values compared to chlorophyll *a* concentrations. $\varepsilon_{fattyacids}$ is the weighted average of
*n*C14:0, *n*C16:1, *n*C16:0, *n*C18:0 fatty acids and the *n*C20:5 PUFA from Jetty samples taken
from August 2010 – December 2011.
**Figure 2**
Phytoplankton diversity and abundance (measured in cells $L^{-1}$) observed in the coastal North
Sea between August 2010 – December 2011.
**Figure 3**
Order-level bacterial diversity and abundance in North Sea water based on the 16S rRNA gene
sequence.
**Figure 4**
Relative abundance of fatty acids and chlorophyll *a* concentration in North Sea SPM. (a)
*n*C14:0, (b) *n*C16:1, (c) *n*C16:0, (d) *n*C18:x, (e) *n*C18:0, (f) *n*C20:5 PUFA and chlorophyll *a*.
**Figure 5**



The D/H fractionation between fatty acids and North Sea water for fatty acids derived from
suspended particulate matter in North Sea water samples. Plotted are the the $\varepsilon_{lipid/water}$ values of
$n$C14:0, $n$C16:1, $n$C16:0, $n$C18:0 fatty acids and $n$C20:5 PUFA. Error bars are the standard
deviation of the duplicate measurements of the fatty acids.
**Table 1**
D/H fractionation between fatty acids and North Sea water for fatty acids derived from
suspended particulate matter in North Sea water samples.





Table 1

| Date | Salinity | δD$_{water}$ [‰] (estimated) | ε$_{lipid/water}$ [‰] | | | | | ε$_{ΣFA}$ [‰] |
| | | | C14:0 | C16:1 | C16:0 | C18:0 | C20:5 PUFA | weighted average C14, C16:1, C16, C18 |
|---|---|---|---|---|---|---|---|---|
| 16/08/10 | 27.3 | -8.2 | -212±0 | -194±1 | -194±2 | -178±3 | -185±1 | -196 |
| 30/08/10 | 29.7 | -4.1 | -218±2 | -198±2 | -186±1 | -182±0 | -195±1 | -197 |
| 15/09/10 | 30 | -3.6 | -213±2 | -203±1 | -194±0 | -183±1 | -177±1 | -201 |
| 28/09/10 | 24.7 | -12.6 | -209±0 | -188±0 | -182±1 | -187±1 | -197±2 | -190 |
| 15/11/10 | 30 | -3.6 | -211±2 | -200±0 | -179±1 | -197±0 | N.D. | -192 |
| 26/11/10 | 24.8 | -12.4 | -216±2 | -192±1 | -178±2 | -193±2 | N.D. | -191 |
| 10/12/10 | 27.1 | -8.5 | -218±0 | -181±0 | -184±0 | -195±0 | N.D. | -191 |
| 17/12/10 | 24.1 | -13.6 | -221±2 | -182±1 | -183±1 | -177±2 | N.D. | -188 |
| 10/01/11 | 27.8 | -7.3 | -215±3 | -195±1 | -180±0 | -198±0 | N.D. | -191 |
| 24/01/11 | 23.0 | -15.5 | -200±2 | -179±0 | -183±0 | -180±1 | -197±2 | -183 |
| 17/02/11 | 29.3 | -4.8 | -219±1 | -204±0 | -191±0 | -203±1 | N.D. | -200 |
| 08/03/11 | 25.8 | -10.7 | -218±6 | -206±2 | -197±1 | -173±4 | -227±8 | -203 |
| 23/03/11 | 26.8 | -9.0 | -234±1 | -209±1 | -198±0 | -182±5 | -234±1 | -208 |
| 05/04/11 | 29.2 | -4.9 | -219±0 | -206±3 | -205±1 | -208±5 | -220±3 | -208 |
| 19/04/11 | 27.7 | -7.5 | -229±0 | -219±1 | -215±0 | N.D. | -235 | -214 |
| 03/05/11 | 31.1 | -1.7 | -237±5 | -224±1 | -213±2 | -210±2 | -235±2 | -223 |
| 18/05/11 | 31.8 | -0.5 | -219±0 | -205±0 | -197±2 | -177±0 | -213±1 | -203 |
| 17/06/11 | 32.0 | 0.7 | -225±2 | -211±0 | -196±3 | -191±0 | N.D. | -206 |
| 30/06/11 | 31.2 | -1.6 | -224±1 | -208±1 | -200±1 | -173±6 | -212 | -209 |
| 15/07/11 | 30.0 | -3.6 | -202±1 | -192±0 | -185±0 | -178±2 | -215±2 | -191 |
| 27/07/11 | 26.3 | -9.9 | -213±3 | -192±3 | -195±0 | -172±0 | -193±6 | -194 |
| 08/08/11 | 29.4 | -4.6 | -219±6 | -198±2 | -197±3 | -176±7 | -231±2 | -200 |
| 22/08/11 | 26.9 | -8.9 | -224±1 | -195±0 | -182±4 | -183±2 | -195 | -196 |
| 06/09/11 | 26.8 | -9.0 | -217±5 | -210±0 | -213±3 | -209±1 | -211±1 | -212 |
| 21/09/11 | 30.1 | -3.4 | -215±0 | -201±0 | -182±0 | -191±1 | N.D. | -193 |
| 11/10/11 | 32.8 | 1.2 | -214±3 | -192±0 | -184±2 | -189±3 | -227 | -192 |
| 28/10/11 | 32.2 | 0.1 | -217±0 | -188±0 | -181±2 | -184±1 | -207±4 | -189 |
| 15/11/11 | 28.9 | -5.5 | -208±12 | -194±2 | -187±4 | -179±6 | -217 | -192 |
| 28/11/11 | 31.7 | -0.7 | -217±0 | -192±0 | -189±1 | -180±2 | -197±1 | -195 |
| 16/12/11 | 31.7 | -0.7 | -198±6 | -179±2 | -173±3 | -187±2 | N.D. | -180 |

$n$C16:1*: double bond at the ω7 position







fig01









fig03






