# Peer review of "Seasonal changes in the D/H ratio of fatty acids of pelagic"

_Biogeosciences, 2016_

## Referee Comment (RC1) · Anonymous Referee #1 · 21 May 2016

Review of Heinzelmann et al. "Seasonal changes in the D/H of fatty acids of pelagic microorganisms in the coastal North Sea" for Biogeosciences

This manuscript provides an interesting environmental test to the hypothesis that changes in the net metabolism of microbial ecosystems are reflected in the hydrogen isotope composition of constituent lipid biomarkers. The authors collected a time series of lipid, DNA, and cell count samples from the North Sea over the course of an annual cycle. The authors do observe changes in both average and compound-specific fractionation and attribute those changes to varying abundances of heterotrophic vs. autotrophic microorganisms. While the dominant photoautotrophs are quantified, no data is presented on the abundance of heterotrophic bacteria. The sequencing data

presented does suggest that heterotrophs are present and their relative abundance varies, but the reader is left to speculate on the true balance of autotrophy and heterotrophy in the system through time. As this is the proposed driver of isotopic variability, this is a problematic omission. In addition, there are substantial problems with the quantification and presentation of errors which appear to significantly underestimate the final uncertainty on isotope values. I suggest revision prior to publication of this manuscript.

Substantial comments: 85: What does 'general' metabolism mean? Perhaps 'Integrated' or 'Net' this is a problem throughout the manuscript. Please define exactly what you mean, then stick to a single term.

300: Is this relative to total fatty acid? I would much rather see these written per liter of sample. With the way it is presented it is impossible to tell if 18:0 went down because 20:5 went up, or because it actually went down. You could show both in a split figure if necessary.

324-327: This is not that strong a correlation and I am suspect you are underestimating the error associated with your water isotope estimates. Just looking that the plot a salinity around 27 could yield dDwater of anywhere from -5 to -10. Please justify the 1.5 permil number and revise as necessary. Also there are no errors shown on the water isotope measurements themselves, but all of the errors need to be propagated to the water isotope estimates. What do the errors on dD of fatty acid measurements mean? Does it include error on internal and external standards? 1 permil would be an exceptionally low error for compound specific isotope measurements... in fact lower than those for individual compounds in the Arndt B standard mix. Both water isotope errors and dD lipid errors need to be propagated into your epsilon values.

346-353: This is an interesting suggestion, but needs better support. What data exists on the use of storage products among algae? How long does and individual alga live (and thus is a seasonal physiological pattern reasonable)? I have heard of dark

reactions becoming a dominant metabolic process on the diurnal, but not seasonal, timescales. What is the resonance time of a 20:5 in a diatom? Its not clear from your reference list which Zhang 2009 is 2009a, but Xinning Zhang's paper is not a good reference here as it does not include new data for phototrophs, metabolic switching, or concurrent metabolisms.

365-368: What do they make instead? Table 6 says that Rhodobacteria make mainly 18:1, 16:0, 16:1, and 18:0, so this statement seems to be both factually and logically inconsistent. If Flavobacteria dominate reads and make all 15:0 and branched 15, why don't you see these lipids? Either the lipid quantification or attribution must be wrong.

388-391: I don't agree with excluding 20:5 from the average. Algae are also contributing to the other pools. Also I am not sure that I agree with the average following the same trends as compound specific measurements. Perhaps plotting the average behind the C-S data would strengthen this statement.

408: This discussion needs to be more quantitative as you have most of the necessary data. The biggest hang up is a lack of data on the true bacterial abundance. I am not sure that you can really speak to the relative importance of autotrophy vs. heterotrophy at a given moment without at least an estimate. The presence of an algal bloom is not enough as we know that heterotrophic populations increase in response.

You do have estimates of relative microbial abundances and estimates of which fatty acids they make—If you estimate the expected FA abundances based on these data, do they match the observed FA pattern? If not does that provide insight into the abundance of heterotrophs? Then since you know the metabolism of each organism (broadly) and therefor have an expectation of fractionation, can you predict what the fractionation should be based on community and evaluate match to the measurements?

Minor comments:
17-18: The running title is just as long as the original title.

46: references are needed after 'effect'

47-53: Be clear through this discussion if you are talking about all FAs, just C16, or an average fractionation.

57-60: This is a little misleading. 'Main' implies the majority whereas NAPDH donates $\sim$50% of H to fatty acid . H2O is the ultimate source of H to autotrophs.

63: I am not sure what 'core' metabolism is in the context.

65: Dawson et al. (2015) recently tested the effect of microbial interaction on D/H.

66-68: Specify that these were targeting marine samples

71-81: The discussion of Osburn et al. 2011 could be more concise. Also, the bigger issue related to the extrapolation of these results more broadly is not that it was a low diversity environment, but rather that is one full of highly unusual (hyper)thermophilic organisms. The Aquificales are not exactly cosmopolitan microbes.

95: Your data do not show an autumn bloom at all.

95-96: Is there data on bacterial abundance from this site? This is necessary complementary information to your phytoplankton counts.

112: What was the sampling depth? Was it consistent between samples? Is turbulence of the water a concern? Some of the changes in bacterial abundance that you note could be influenced by incorporation of sediment into the sample.

117: 0.7 is not very small compared to many marine heterotrophic bacteria. The clogging noted in 135-138 is also troubling. Was any microscopy performed to confirm what was actually analyzed? How can you be sure the bacterial population was captured?

186: Strange formatting on your instrumentation abbreviation protocol throughout 2.3. 'elemental analysis/TC/irMS (EA/TC/irMS)' is rather repetitive

189-195: Cyanobacteria are not usually considered algae. I find it strange that you counted the cyanobacteria, but did not do a total cell count.

242-244: This belongs in the methods.

247: 4 is not particularly low given the range you present. What is the error on these measurements? The graph seems to show a trend during this time period rather than a constant low.

258: Microbial Diversity would include that of the Archaea. You discuss only bacterial diversity.

289: The 'Other' group makes up a large percentage of your samples, You should add a section documenting any taxa that demonstrate seasonable variability.

335: Misleading. At times this is among the most D-enriched compound.

337-339: It becomes unclear at times if you are discussing your own data or that from the literature. I believe this is the literature and therefor requires references. Try to clarify this throughout the discussion.

342-344: But why would Phaeocystis produced enriched lipids? Any why does it matter of 20:5 is a trace or major constituent? To my knowledge there is no published information relating the relative abundance of a lipid to its relative D-enrichment or D-depletion.

354: I think on average it is the most D-depleted

372-376: The table needs to be reformatted in order to make these more general conclusions.

386: This is not evident for 18:0

392-394: Can the correlation of trends be shown statistically?

418-419: This could be said for fatty acids from almost all environments.

422: Except that 20:5 also becomes very enriched and is derived from autotrophs.

Figure 1: Error bars on epsilon measurements (propagated as discussed above). If you come up with a measure of heterotrophic abundance, I would like to see it plotted here as well.

Figure 3: The 'Other' category is way to big. The usual standard is to lump phyla that comprise <1% of the total. This could be pushed to maybe 3 or 5% here since the trace constituents probably are not important. I would be particularly interested in any phyla with transient high abundances.

Figure 4 and 5: It would be helpful to have some shaded bars delineating seasons to guide the eye between these subplots. Table S4: Please report concentrations instead of relative abundance. Table S6: This would be far more useful reformatted in either tabular or heatmap form. It should be possible to scan down a column for a single FA and identify those that make it and those that don't. For the abundances that you discuss in the text, the original reference should be cited rather than the table itself. Figures S1-S3: I am not sure these trees are necessary. Figure S4: Error bars on each input measurement. As noted above you need to include more information on the error associate with the extrapolated water isotope values. Perhaps confidence intervals on this figure would be useful in that context. It would be also useful to color code these points by date to illustrate seasonal trends in salinity and water isotopes that might be underlying the organic isotope trends.

---

## Referee Comment (RC2) · Anonymous Referee #2 · 24 May 2016

Review of Heinzelmann et al. Seasonal changes in the D/H ratio of fatty acids of pelagic microorganisms in the coastal North Sea

This ms seeks to show the applicability of lipid D/H as a biomarker for microbial metabolism in the environment by correlating lipid D/H with large metabolic shifts expected to be observed within a coastal plankton community over a bloom. The authors hypothesize that lipid water fractionation should shift to reflect the transition from an ecosystem dominated by photoautotrophy, during the spring bloom, to one where bacterial heterotrophy becomes progressively more important post-bloom. In addition to lipid D/H and water D/H measurements, the ms presents results from DNA sequencing of the bacterial community.

The bloom scenario provides a very interesting test of the usefulness of environmental lipid D/H for metabolism but the study hinges on showing the balance of autotrophic vs heterotrophic biomass in the system, which has not been clearly demonstrated. DNA sequencing data can not address the contribution of heterotrophic bacteria to the lipid pool relative to autotrophs, most of which are algae. Could any NPP, respiration measurements be used to show that the community shifts towards heterotrophy ? Additionally, the overall shift in lipid water fractionation is small (30 per mil increase in the weighted fractionation), and while it is in the direction that is consistent with a growing contribution of heterotrophy post bloom, the value of the most positive measurement (-170 permil) does not fall in the range of fractionation values that clearly correspond to heterotrophy (>-150, conservatively >-100 permil ). Without knowing more about the contributions of autotrophy vs heterotrophy over the season, it does not seem possible to distinguish whether enriched plankton fatty acids reflect increases in heterotrophic contribution to lipids or a shift in the autotrophic population or its metabolism. Regarding shift in phytoplankton metabolism, how can the variation in the fractionation of the algal fatty acid from -240 to -180 permil be explained ?

Since the relative importance of autotrophy versus heterotrophy has not been clearly demonstrated and the change in lipid water fractionation is relatively small, I believe that the conclusions are overstated and suggest that a significant revision be made before resubmission.

Specific comments:

Lines 29-30: add fractionation values for algal and C18:0 lipids

Lines 30: there is > 50 per mil variability in C20:5 fatty acid, as much variability as in the weighted average . . .

Line 36: there is no data that directly addresses the contribution of heterotrophs vs autotrophs to fatty acid pool

Line 37: can be a useful when combined with other measurements

Line 46. Reference

Line 58-59: there is a lot that needs to be verified regarding dD of NADPH ... suggest these are hypotheses

Line 63: not exactly sure what is meant by "integrated" core metabolism – how would you define this? Is there any need to differentiate between integrated and specific metabolisms? I guess authors are hinting at the fact that lipid D/H can not provide much information on specific microbial groups if lipids are not group specific. In any case, does integrated mean the amount of carbon or some element cycling through a particular metabolism or biomass from a microbial group using a metabolism?

Lines 61-81: Osburn et al were the first to explore the environmental applicability of lipid D/H using hot spring systems where a wide range of metabolic groups are physically separated. Lipid D/H over the course of a phytoplankton bloom is another interesting environmental test that extend lipid D/H to more widespread systems– I think there is no need to bring up the relative diversity of the systems. And if so, a reference is needed to compare the diversity of hot springs with plankton communities.

Line 85: again the idea of general metabolism is very vague – in this context it would be the balance of autotrophy vs heterotrophy?

Lines 99-109: Do the bacteria synthesize fatty acids de novo or utilize the lipids of the autotrophs? If there is research here it should be mentioned in the intro and included in the interpretation of the results.

Line 329: does it make sense to be so precise with the fractionation values given the error of D/H is a few per mil?

Line 335 -336: need to acknowledge significant variability in C20:5 PUFA, yes it's one of the most depleted, but it also gets relatively enriched, within the range of fractionation expressed by heterotrophs

Line 337, 339: provide references

Line 343: reference for diatom lipid profile

Line 348: variations in the dD of NADPH varying with different pathways of production remains a hypothesis

Line 374: If 18:0 is mainly derived from heterotrophic bacteria, wouldn't the fractionation be expected to more positive (> - 150 per mil)? Many of the 18:0 values are below -190 per mil, and there is also significant variability between points (e.g. Dec through March).

Line 380: I believe there is some data from Valentine 2002, an H2 +CO2 acetogen?

Line 403-404: Please address the possibility of a shift in the photoautrophic population as explaining the ~30 per mil shift upwards

Line 411: references beyond Heinzelmann 2015 are needed.

Tables and Figure:

Table 1. Possible to include relative abundance of differenty fatty acids (parentheses after isotope values)?

Figure 1. include C20:5 pufa as "calibrating" data for autotrophy

Fig 2: Do you have total cell counts? If so, maybe you can use this to get a rough estimate of the heterotroph cell abundance.

Figure 4,5: add in bar highlighting bloom data, the most obvious trends in fractionation are the dips in C16:0 data, one corresponding to the bloom, and another dip in October. The fractionation goes as low as the bloom fractionation but is unrelated to photoautotrophy?

---

## Referee Comment (RC3) · A. Sessions (Referee) · 28 May 2016

The purpose of this manuscript is to demonstrate a seasonal shift in the $\delta$D values of marine fatty acids; such a pattern has not previously been reported. This manuscript is concise and well-written, and was a pleasure to read; the relative brevity belies a significant amount of effort that must have gone into the analysis of samples. The data all appear to be of high quality.

The main conclusion is that the $\delta$D value of average fatty acids gets more negative during the spring bloom; authors attribute this pattern to an increase in autotrophic contributions to particulate biomass during the spring, and greater heterotrophic input later in the summer. The attribution of this pattern to changing auto/heterotrophic inputs

is very interesting, and if it proves to be correct would be an exciting demonstration of the utility of lipid $\delta$D values as ecological tracers. So I think the work is definitely significant. However, I don't think the case is as firmly settled as the current manuscript makes it appear. The main issue I have is that all of the fatty acids – including 20:5, which is a putative algal marker – show the same temporal pattern of depletion then enrichment over the year. Plus the use of 3um filters should preferentially collect algae rather than bacteria. So it seems entirely possible that the observed pattern is due (mainly or entirely) to changing fractionations by phytoplankton, rather than changing relative inputs from bacteria and algae. Perhaps the authors can think of some way to rule this hypothesis out; if not, I think it has to be presented as an alternative hypothesis that could explain the data.

Other minor comments:

L56-57; its the hydrogen supplied by NADPH (not the whole molecule) that matters, and its the reduction of NADP+ to NADPH that matters, not its biosynthesis. I'm sure the authors know this, just be precise in the wording.

L91: 'comprised of' rather than 'formed by'

L137-138: I agree that the 3um filters can trap some bacteria as they get loaded, but still your sample will be strongly biased towards algae rather than bacteria. You might want to point this out, because the seasonal signal you observed could be even larger in a sample that more effectively traps bacterial biomass. Collecting bigger samples on smaller pore-size filters would also allow to test your assumption that bacterial lipids are D-enriched.

L147. Boiling point of methanol is around 65C, so how did you reflux something at 140C in methanol? Would have to be under pressure...?

L183: I believe the internal standard (squalane) is allowing you to test accuracy, rather than precision. Precision is simply the reproducibility of unknown values.

L261-275. This is the one place where I thought the manuscript could be a bit shorter. The patterns of changing abundance are shown clearly in the figure, and not every detail needs to be recounted here in the text. Just point out the most important trends.

L309-311. I believe a similar trend (shortest-chain FA are most D-depleted) has been observed in many pure cultures and single (macro)organisms as well. Here it would be worth comparing the trend you see to some previously published results. Later, when you try to attribute C18:x to greater heterotrophic input, I would be cautious: the fact that it is D-enriched relative to 14:0 might be true even in algae, and would not necessarily require heterotrophic input. You could also point out that a similar pattern (increasing $\delta$D with chain length, except for PUFA's) has been observed in other environmental samples, like Maggie's Yellowstone paper and Ashley Jones' Santa Barbara paper.

L347. I agree this is theoretically possible, but the timing is weird, because its in the summer (not fall). Why would plankton switch to more storage products in the middle of summer, when they have the most sunlight. Even if they are nutrient-limited they can still make sugars...

L354-355 This is where I'd be cautious; similar pattern seen in many individual organisms, I think.

L375. The statement that "C18:0 will be mainly derived from heterotrophic bacteria" is not supported by table S6, which just lists the relative abundance of fatty acids. If algal inputs are 3 orders of magnitude greater than bacteria, then the bulk C18:0 will still be mainly from the algae, even though bacteria have a higher relative abundance. Note that I am not saying it is not possible for C18 to be mainly bacterial, just that the data presented does not prove that point.

L377-378. I am currently working on a manuscript with Dave Valentine that compiles published fractionations for a variety of chemoautotrophs, and shows that they are not statistically significantly different from photoautotrophs. The first few organisms that

were studied, that yielded such huge depletions, turn out to be anomalous. So I would probably just delete this sentence.

L380: You are comparing epsilon values (from current study) to the $\delta$D values from Sandra's earlier culture work. The epsilon values from her study for chemoautotrophs are -217 to -275‰ (numbers taken from the abstract).

L393. I think you mean Figure 1 here? Fig 5 does not show data for the summed fatty acids.

Nice job!

Alex Sessions

---

## Author Comment (AC1) · 14 Jul 2016

Reviewer 1 This manuscript provides an interesting environmental test to the hypothesis that changes in the net metabolism of microbial ecosystems are reflected in the hydrogen isotope composition of constituent lipid biomarkers. The authors collected a time series of lipid, DNA, and cell count samples from the North Sea over the course of an annual cycle. The authors do observe changes in both average and compound-specific fractionation and attribute those changes to varying abundances of heterotrophic vs. autotrophic microorganisms. While the dominant photoautotrophs are quantified, no data is presented on the abundance of heterotrophic bacteria. The sequencing data presented does suggest that heterotrophs are present and their relative abundance varies, but the reader is left to speculate on the true balance of autotrophy and heterotrophy in the system through time. As this is the proposed driver of isotopic variability, this is a problematic omission. In addition, there are substantial problems with the quantification and presentation of errors which appear to significantly underestimate the final uncertainty on isotope values. I suggest revision prior to publication of this manuscript.

We want to thank the reviewer for her/his constructive comments. We are aware that unfortunately we do not have data about the actual abundance of heterotrophic bacteria in our time series. While we can say something about the total abundance of photoautotrophs and the relative contribution of different (heterotrophic) bacteria species to the bacterial fatty acid pool, the actual contribution of the bacteria to the total fatty acid pool is left to speculation. However, the NIOZ Jetty system has been subjugated to intensive sampling and study over the past decades and its microbial population dynamics at the jetty is therefore well known. Furthermore, it has been shown that the system is quite seasonally stable throughout the whole time it has been studied. Brandsma et al. 2012 for example studied both the phytoplankton and bacterial community from 2007 to 2008. Their study included cell counts for algae, cyanobacteria and bacteria. They showed that the cell number of algae increased during spring forming the known spring bloom, the cell number of cyanobacteria was in general low except for late summer/early autumn. This fits with our observation. At the same time they showed that the total cell number of the bacteria was ten times as high as for the diverse phytoplankton. Additionally, they showed that bacterial cell numbers were lowest during the spring bloom and increased right after. During the rest of their sampling period this cell number stayed more or less constant. This suggested that the microbial community was dominated by the heterotrophic bacterioplankton following the spring bloom. While especially the autumn bloom weakened over the last years the general shift in the microbial community most likely stayed the same. Due to the stability of this systems throughout the last decades we expect similar dynamics during our sampling period and are therefore confident that the shift between photoautotrophy and heterotrophy is

the driver behind changes in the hydrogen isotopic composition of fatty acids sampled throughout our sampling period. We will include this background information in the manuscript to make things more clear. Concerning the quantification and presentation of errors, we will revise this in the manuscript.

Substantial comments: 85: What does 'general' metabolism mean? Perhaps 'Integrated' or 'Net' this is a problem throughout the manuscript. Please define exactly what you mean, then stick to a single term.

With general or core metabolism we mean photoautotrophic, chemoautotrophic or heterotrophic metabolism. We will clarify this in the manuscript.

300: Is this relative to total fatty acid? I would much rather see these written per liter of sample. With the way it is presented it is impossible to tell if 18:0 went down because 20:5 went up, or because it actually went down. You could show both in a split figure if necessary.

The relative abundance of the fatty acids is indeed relative to the total fatty acid that are mentioned in the manuscript. Since the hydrogen isotopic composition of the fatty acids also reflects the relative contributions from different metabolisms absolute amounts or fatty acid concentrations are not relevant for the main focus of this manuscript.

324-327: This is not that strong a correlation and I am suspect you are underestimating the error associated with your water isotope estimates. Just looking that the plot a salinity around 27 could yield dDwater of anywhere from -5 to -10. Please justify the 1.5 permil number and revise as necessary. Also there are no errors shown on the water isotope measurements themselves, but all of the errors need to be propagated to the water isotope estimates. What do the errors on dD of fatty acid measurements mean? Does it include error on internal and external standards? 1 permil would be an exceptionally low error for compound specific isotope measurements: : : in fact lower than those for individual compounds in the Arndt B standard mix. Both water isotope errors and dD lipid errors need to be propagated into your epsilon values.

We agree with the reviewer that the propagated error including the salinity water isotope error on the $\varepsilon$ values need to be calculated and will be revised in the manuscript, however the observed trend in $\varepsilon$ versus time is already present in the hydrogen isotopic composition of the fatty acids, a value that is not affected by the uncertainty in the salinity vs. water $\delta$D value. Therefore we think that our discussion is correct and our conclusions can be drawn even though there is scatter in the salinity $\delta$D water relationship.

346-353: This is an interesting suggestion, but needs better support. What data exists on the use of storage products among algae? How long does and individual alga live (and thus is a seasonal physiological pattern reasonable)? I have heard of dark reactions becoming a dominant metabolic process on the diurnal, but not seasonal, timescales. What is the resonance time of a 20:5 in a diatom? Its not clear from your reference list which Zhang 2009 is 2009a, but Xinning Zhang's paper is not a good reference here as it does not include new data for phototrophs, metabolic switching, or concurrent metabolisms.

The utilization of storage products by algae during nutrient limitation is a possible explanation for changes in the hydrogen isotopic composition of the algae biomarker fatty acid C20:5 PUFA. We will make clear in the manuscript that this is a hypothesis. It is also possible that the situation with the spring bloom behaves comparable to that of a batch culture in the sense that with increasing nutrient limitation more and more cells will be in "stationary" growth phase and later even in the declining or death phase. It has been previously shown that the fatty acids of algae became enriched in D with increasing age of the culture. This factor might also play a role in the fact that fatty acids derived from the same organisms become enriched following the phytoplankton bloom.

365-368: What do they make instead? Table 6 says that Rhodobacteria make mainly 18:1, 16:0, 16:1, and 18:0, so this statement seems to be both factually and logically inconsistent. If Flavobacteria dominate reads and make all 15:0 and branched 15, why

don't you see these lipids? Either the lipid quantification or attribution must be wrong.

We apologize for the mistake in Table S6, Rhodobacteria mainly produce C18:1 and only, in comparison, minor amounts of C16:0 and C18:0. We will correct this error in the revised manuscript.

388-391: I don't agree with excluding 20:5 from the average. Algae are also contributing to the other pools. Also I am not sure that I agree with the average following the same trends as compound specific measurements. Perhaps plotting the average behind the C-S data would strengthen this statement. We excluded the C20:5 PUFA from the average since we wanted to present an average of the general fatty acids without a specific source in order to have a more unbiased look at the whole community. However we also calculated the average including the C20:5 PUFA and the values only differ between 1 to 4 ‰ from the values excluding the C20:5 PUFA. Therefore, we don't think that it is necessary to include the C20:5 PUFA in the average. We agree with the reviewer that potentially in the future the approach might be to look at general fatty acids and specific biomarkers for different metabolisms to account for contributions from different organisms, investigating what works and what doesn't is exactly the purpose of this study.

408: This discussion needs to be more quantitative as you have most of the necessary data. The biggest hang up is a lack of data on the true bacterial abundance. I am not sure that you can really speak to the relative importance of autotrophy vs. heterotrophy at a given moment without at least an estimate. The presence of an algal bloom is not enough as we know that heterotrophic populations increase in response.

As we mentioned above, Brandsma et al. 2012 showed changes in the total number of bacteria and phytoplankton cells with time, with the bacterial abundance being lowest during the spring bloom. We are therefore confident that during our sampling period similar changes happened.

You do have estimates of relative microbial abundances and estimates of which fatty

acids they makeâËŸAËĞ TIf you estimate the expected FA abundances based on these data, do they match the observed FA pattern? If not does that provide insight into the abundance of heterotrophs? Then since you know the metabolism of each organism (broadly) and therefor have an expectation of fractionation, can you predict what the fractionation should be based on community and evaluate match to the measurements?

This would be an interesting exercise. Unfortunately, absolute amounts of fatty acids produced by the various bacteria are also subject to growth conditions and potentially species, strain or eco type specific. The strains and species analysed grown in pure culture and under optimal conditions might not reflect what is going on in nature. Therefore it won't be possible to make any precise calculations.

Minor comments: 17-18: The running title is just as long as the original title.

We will shorten the running title as requested.

46: references are needed after 'effect'

We will add a reference.

47-53: Be clear through this discussion if you are talking about all FAs, just C16, or an average fractionation.

Thank you for the remark. We will make it clear in the manuscript in order to avoid any misunderstandings.

57-60: This is a little misleading. 'Main' implies the majority whereas NAPDH donates 50% of H to fatty acid . H2O is the ultimate source of H to autotrophs.

We will rephrase it in the revised manuscript.

63: I am not sure what 'core' metabolism is in the context.

As mentioned above we mean with general or core metabolism photoautotrophic,

chemoautotrophic or heterotrophic metabolism. We will clarify this in the manuscript.

65: Dawson et al. (2015) recently tested the effect of microbial interaction on D/H.

We appreciate the comment from the review and apologize for not mentioning Dawson et al.. It will be changed in the manuscript.

66-68: Specify that these were targeting marine samples

We will specify this in the manuscript.

71-81: The discussion of Osburn et al. 2011 could be more concise. Also, the bigger issue related to the extrapolation of these results more broadly is not that it was a low diversity environment, but rather that is one full of highly unusual (hyper)thermophilic organisms. The Aquificales are not exactly cosmopolitan microbes.

We thank the reviewer for the remark and we will shorten this part of the introduction and point out that the diversity of this study site is rather unusual.

95: Your data do not show an autumn bloom at all.

As mentioned in the introduction Philippart et al. 2010 showed that the autumn bloom weakened over the last years. Thus it is not surprising that the autumn bloom might not be so obvious in our data.

95-96: Is there data on bacterial abundance from this site? This is necessary complementary information to your phytoplankton counts.

We understand that this might be one of the major limitations of our data set, but unfortunately there is no data on bacterial abundance. Nevertheless, we think that our interpretation of the data correct especially taking all the previous work at this site into account, but we will point out other possibilities as requested from the reviewers.

112: What was the sampling depth? Was it consistent between samples? Is turbulence of the water a concern? Some of the changes in bacterial abundance that you note

could be influenced by incorporation of sediment into the sample.

The samples were surface samples (depth max 50 cm below surface) and always taken during high tide. Therefore, the water is considered to be North Sea water. While we cannot rule out that turbulences might have led to an input of the sediment to the samples, the sample site is next to an artificial dyke which is stabilized by basalt and concrete and therefore the amount of sediment at the surface is likely limited. This is also evident by the clear seasonality in our signals which would be lost if we had a dominant sediment input. We will provide a more detailed description in the revised text.

117: 0.7 is not very small compared to many marine heterotrophic bacteria. The clogging noted in 135-138 is also troubling. Was any microscopy performed to confirm what was actually analyzed? How can you be sure the bacterial population was captured?

We are aware that there are bacteria which will be smaller than the 0.7 $\mu$m and will therefore not be captured by the filters we used. Unfortunately, for lipid analysis the filters used have to be plastic free and organic solvent resistant and at the time of sampling there were no suitable filters available with a smaller pore size. The clogging of the 3 $\mu$m and the rather low fatty acid yield of the 0.7 $\mu$m filters lead to the assumption that we captured the majority of the biomass in the water samples. Pitcher et al. 2011 (Limnology and Oceanography) also showed in their work a good signal for the small Thaumarchaeota in 3 $\mu$m filters. Of course we cannot rule out that we still have a discrimination towards bigger cells and that a portion of the smaller cells might not have been captured and therefore analysed. Unfortunately, at the moment there is no methodological possibility to change those biases.

186: Strange formatting on your instrumentation abbreviation protocol throughout 2.3. 'elemental analysis/TC/irMS (EA/TC/irMS)' is rather repetitive

We will change that in the manuscript.

[Figure]

189-195: Cyanobacteria are not usually considered algae. I find it strange that you counted the cyanobacteria, but did not do a total cell count.

We will change algae to photoautotrophic microorganisms in the manuscript.

242-244: This belongs in the methods.

We will move this sentence to the method section.

247: 4 is not particularly low given the range you present. What is the error on these measurements? The graph seems to show a trend during this time period rather than a constant low.

Unfortunately the exact error for these measurements is not known, but the reliability of the HPLC approach has been previously described in Philippart et al. 2010.

258: Microbial Diversity would include that of the Archaea. You discuss only bacterial diversity.

We will change the header in the manuscript.

289: The 'Other' group makes up a large percentage of your samples, You should add a section documenting any taxa that demonstrate seasonable variability.

Our figure shows the bacterial diversity on an order level. These orders contributed to at least 3% of the reads with at least one point during sampling. The other group consists of all the other bacterial orders which at no point in sampling contribute to at least 3% of the total bacteria reads. There was only small seasonal variability seen in these orders. Nevertheless, we will add a table to the supplementary with the relative contribution to the reads on a phyla level because that would also include orders which contribute less than 3% to the total bacterial reads.

335: Misleading. At times this is among the most D-enriched compound.

In general, the C20:5 is the most depleted fatty acid. We will clarify this in the text.

[Figure]

337-339: It becomes unclear at times if you are discussing your own data or that from the literature. I believe this is the literature and therefor requires references. Try to clarify this throughout the discussion.

We thank the reviewer for pointing out this and will add references when necessary throughout the discussion.

342-344: But why would Phaeocystis produced enriched lipids? Any why does it matter of 20:5 is a trace or major constituent? To my knowledge there is no published information relating the relative abundance of a lipid to its relative D-enrichment or D-depletion.

The same fatty acid will have a different isotopic value depending on the source organism, even when both express the same metabolism. We do not know the actual value for the C20:5 PUFA produced either by diatoms or Phaeocystis, however changes in the abundance of either species will have an effect on the isotopic value of the fatty acid in the samples. Therefore, it is possible that changes in the isotopic value are due to changes in the abundance in the different source organisms. Additionally, Phaeocystis is a colony forming algae which potentially affects their metabolism, the type of lipids they produce, the recycling of organic matter, the production of osmolytes, exchange of water over the cell membrane and potentially different water pools, other than just North Sea water, being used during fatty acid and NADPH synthesis (the colonies form a "ball" around an internal water pool). All these factors could potentially affect the hydrogen isotopic composition of an individual lipid, on top of the integrated metabolism signal. At lower concentrations of a specific fatty acid the contribution from a single source will be more easily picked up than at higher concentrations.

354: I think on average it is the most D-depleted

Indeed, the C14:0 fatty acid seems to be one of the most depleted fatty acid.

372-376: The table needs to be reformatted in order to make these more general

conclusions.

It is unclear to us what the reviewer exactly means.

386: This is not evident for 18:0

We will change that in the manuscript to most fatty acids.

392-394: Can the correlation of trends be shown statistically?

As mentioned in the manuscript, figure 1 shows that the chlorophyll a concentration and the $\varepsilon$ value s of the weighted averages follow opposing trends. Additionally, the chlorophyll a measurements have not always been at the same time points as the isotopic measurements. Therefore, a statistical correlation would require an extrapolation for the chlorophyll a data, which would be rather imprecise. So, it is impossible to add a statistical significance.

418-419: This could be said for fatty acids from almost all environments.

This is indeed true, however in this study we only look at our sample set and don't necessarily compare it with other studies.

422: Except that 20:5 also becomes very enriched and is derived from autotrophs.

As discussed above the PUFA has multiple sources that might changes during the course of a bloom as well as potential "growth phase" changes of the organisms present during the course of a bloom also resulting in changes in hydrogen isotope fractionation. Additionally, photoautotrophic microorganisms have been shown to produce fatty acids with $\varepsilon$ values ranging between -150 and -250 ‰ The isotope values of the C20:5 PUFA clearly fall within this range.

Figure 1: Error bars on epsilon measurements (propagated as discussed above). If you come up with a measure of heterotrophic abundance, I would like to see it plotted here as well.

We will add the error bars to the graph.

Figure 3: The 'Other' category is way to big. The usual standard is to lump phyla that comprise <1% of the total. This could be pushed to maybe 3 or 5% here since the trace constituents probably are not important. I would be particularly interested in any phyla with transient high abundances.

The Other category contains all microorganisms on an order level which contribute less than 3 % to the total bacteria reads. We will add an additional table for the contribution on a phylum level.

Figure 4 and 5: It would be helpful to have some shaded bars delineating seasons to guide the eye between these subplots.

We will add this to the graphs.

Table S4: Please report concentrations instead of relative abundance. Table S6: This would be far more useful reformatted in either tabular or heatmap form. It should be possible to scan down a column for a single FA and identify those that make it and those that don't. For the abundances that you discuss in the text, the original reference should be cited rather than the table itself.

Since we did not use an internal standard we cannot calculate the actual concentration of the fatty acids.

Figures S1-S3: I am not sure these trees are necessary.

The trees show to which bacterial genera our sequences are related. This allows us to see what the most likely fatty acid compositions of the bacteria in our samples are and is we can associate the isotopic signal of the individual fatty acids to the specific metabolic potential of these genera. The figures are supplied as supplemental information and do not form the core of the manuscript, only those readers who are interested in this type of information can look at it. Therefore, we prefer to keep these supplemental figures.

Figure S4: Error bars on each input measurement. As noted above you need to include more information on the error associate with the extrapolated water isotope values. Perhaps confidence intervals on this figure would be useful in that context. It would be also useful to color code these points by date to illustrate seasonal trends in salinity and water isotopes that might be underlying the organic isotope trends.

As mentioned above we agree with the reviewer that the propagated error including the salinity water isotope error on the $\varepsilon$ values need to be calculated and will be revised in the manuscript, however the observed trend in $\varepsilon$ versus time is already present in the hydrogen isotopic composition of the fatty acids, a value that is not affected by the uncertainty in the salinity vs. water $\delta$D value. Therefore we think that our discussion is correct and our conclusions can be drawn even though there is some scatter in the salinity $\delta$D water relationship. Additionally, the samples were taken during a rather short timescale and it is unlikely that seasonal influences would lead to changes in the salinity/$\delta$D relationship of the water.

---

## Author Comment (AC2) · 14 Jul 2016

Reviewer 2

This ms seeks to show the applicability of lipid D/H as a biomarker for microbial metabolism in the environment by correlating lipid D/H with large metabolic shifts expected to be observed within a coastal plankton community over a bloom. The authors hypothesize that lipid water fractionation should shift to reflect the transition from an ecosystem dominated by photoautotrophy, during the spring bloom, to one where bacterial heterotrophy becomes progressively more important post-bloom. In addition to lipid D/H and water D/H measurements, the ms presents results from DNA sequencing of the bacterial community. The bloom scenario provides a very interesting test of the

usefulness of environmental lipid D/H for metabolism but the study hinges on showing the balance of autotrophic vs heterotrophic biomass in the system, which has not been clearly demonstrated. DNA sequencing data can not address the contribution of heterotrophic bacteria to the lipid pool relative to autotrophs, most of which are algae. Could any NPP, respiration measurements be used to show that the community shifts towards heterotrophy ? Additionally, the overall shift in lipid water fractionation is small (30 per mil increase in the weighted fractionation), and while it is in the direction that is consistent with a growing contribution of heterotrophy post bloom, the value of the most positive measurement (-170 permil) does not fall in the range of fractionation values that clearly correspond to heterotrophy (>-150, conservatively >-100 permil ). Without knowing more about the contributions of autotrophy vs heterotrophy over the season, it does not seem possible to distinguish whether enriched plankton fatty acids reflect increases in heterotrophic contribution to lipids or a shift in the autotrophic population or its metabolism. Regarding shift in phytoplankton metabolism, how can the variation in the fractionation of the algal fatty acid from -240 to -180 permil be explained ? Since the relative importance of autotrophy versus heterotrophy has not been clearly demonstrated and the change in lipid water fractionation is relatively small, I believe that the conclusions are overstated and suggest that a significant revision be made before resubmission. We thank the reviewer for her/his comments and hope that we will be able to address them to her/his satisfaction.

Unfortunately our samples do not allow for any measurements of net primary production or respiration. However, as explained in the rebuttal to the comments of Reviewer 1 we are confident that this environment is indeed driven by a shift from photoautotrophy during the phytoplankton bloom towards heterotrophy following the phytoplankton bloom. This has been shown in a range of previous studies (as mentioned in the manuscript) over a time period of several decades. Additionally, Alderkamp et al. (2006) showed an increase in activity of the heterotrophic bacteria following the decline of the phytoplankton bloom in spring via MICRO-CARD-FISH. We are aware of the different isotopic ranges associated with the individual metabolism types. However, these

ranges were established for pure cultures grown under specific conditions exhibiting only one type of metabolism at the time. In the environment all microorganisms will contribute to the fatty acid pool making the isotopic signal of the fatty acids a reflection of the relative contribution of specific microorganisms to the individual fatty acid pools. Therefore, the mixed environment it can't be expected that the isotopic signals will fall into those quite clear ranges, but will rather fall into a range between those ranges. Changes in the relative contribution of different metabolism types will lead to a shift towards more negative or positive isotopic values while the values will stay in the same range (unless a complete change of metabolism with no input from other metabolism types will happen). Hence we don't think that the changes are too small to make a prediction about shifts in the community metabolism. However, we do agree that for future applications of the hydrogen isotopic composition of fatty acids as community metabolism indicator it might be helpful to look at both general and biomarker fatty acids, reflecting only one type of metabolism. Concerning the wide range of isotopic values of the C20:5 PUFA, as mentioned to reviewer 1 above the PUFA has multiple sources that might changes during the course of a bloom as well as potential "growth phase" changes of the organisms present during the course of a bloom also resulting in changes in hydrogen isotope fractionation.

Specific comments: Lines 29-30: add fractionation values for algal and C18:0 lipids

We will add the requested values to the manuscript.

Lines 30: there is > 50 per mil variability in C20:5 fatty acid, as much variability as in the weighted average : : :

We will add to the sentence to show that the C20:5 is mainly but not always highly depleted.

Line 36: there is no data that directly addresses the contribution of heterotrophs vs autotrophs to fatty acid pool

We will rephrase the sentence in question in the manuscript.

Line 37: can be a useful when combined with other measurements

Thanks for pointing this out, we added this to the sentence.

Line 46. Reference

A reference has been added to the manusript.

Line 58-59: there is a lot that needs to be verified regarding dD of NADPH ... suggest these are hypotheses

We will change this in the manuscript in order to point out that these are just hypotheses.

Line 63: not exactly sure what is meant by "integrated" core metabolism – how would you define this? Is there any need to differentiate between integrated and specific metabolisms? I guess authors are hinting at the fact that lipid D/H can not provide much information on specific microbial groups if lipids are not group specific. In any case, does integrated mean the amount of carbon or some element cycling through a particular metabolism or biomass from a microbial group using a metabolism?

Integrated metabolism means all different metabolism types in a community looked at as one.

Lines 61-81: Osburn et al were the first to explore the environmental applicability of lipid D/H using hot spring systems where a wide range of metabolic groups are physically separated. Lipid D/H over the course of a phytoplankton bloom is another interesting environmental test that extend lipid D/H to more widespread systems– I think there is no need to bring up the relative diversity of the systems. And if so, a reference is needed to compare the diversity of hot springs with plankton communities.

We thank the reviewer for her/his remark. We will shorten this part as pointed out.

[Figure]

Line 85: again the idea of general metabolism is very vague – in this context it would be the balance of autotrophy vs heterotrophy?

We will make this more clear in the revised manuscript.

Lines 99-109: Do the bacteria synthesize fatty acids de novo or utilize the lipids of the autotrophs? If there is research here it should be mentioned in the intro and included in the interpretation of the results.

For our study site it is to the best of our knowledge not known if the heterotrophic bacteria do synthesise the fatty acids de novo or if they utilize the lipids derived from the autotrophs.

Line 329: does it make sense to be so precise with the fractionation values given the error of D/H is a few per mil?

We thank the reviewer for the remark, however it is unclear to us what the reviewer exactly has in mind. The precision of the analysis is approximately 3 ‰ however our measurement error is much lower. We agree that reporting hydrogen isotope values with decimal places might not be meaningful.

Line 335 -336: need to acknowledge significant variability in C20:5 PUFA, yes it's one of the most depleted, but it also gets relatively enriched, within the range of fractionation expressed by heterotrophs

Concerning the wide range of isotopic values of the C20:5 PUFA, as mentioned to reviewer 1 above the PUFA has multiple sources that might changes during the course of a bloom as well as potential "growth phase" changes of the organisms present during the course of a bloom also resulting in changes in hydrogen isotope fractionation. Additionally, according to published data the $\varepsilon$ values of the C20:5 PUFA fall within the range associated with photoautotrophic growth (-150 to -250 ‰. Nevertheless, we will mention the variability in the manuscript.

Line 337, 339: provide references

We added the required references.

Line 343: reference for diatom lipid profile

The references are in Table S6.

Line 348: variations in the dD of NADPH varying with different pathways of production remains a hypothesis

It indeed remains a hypothesis at the moment. We will make that clear in the text.

Line 374: If 18:0 is mainly derived from heterotrophic bacteria, wouldn't the fractionation be expected to more positive (> - 150 per mil)? Many of the 18:0 values are below -190 per mil, and there is also significant variability between points (e.g. Dec through March).

While we assume that the C18:0 will be mainly produced by heterotrophic bacteria, diatoms and Phaeocysitis produce this fatty acid in minor amounts. This might as well contribute to the relatively low isotopic values.

Line 380: I believe there is some data from Valentine 2002, an H2 +CO2 acetogen?

Here we meant the summary for all data published in previous papers but summarised in Heinzelmann et al. 2015b. However, we will add the Valentine paper also as reference.

Line 403-404: Please address the possibility of a shift in the photoautrophic population as explaining the 30 per mil shift upwards

Data published by Brandsma et al. 2012 show changes in the total cell number of bacteria and phytoplankton during their sampling period. Although there are changes in the phytoplankton population from Phaeocystis dominated to diatom/cyanobacteria dominated, we think that the shift in the isotopic values is rather due to changes in the relative contribution of heterotrophic bacteria versus photoautotrophic phytoplankton. At the moment published isotopic data shows that changes in the metabolism

leads to bigger shifts in the isotopic composition of fatty acids while different organisms expressing the same metabolism show similar isotopic values.

Line 411: references beyond Heinzelmann 2015 are needed.

We will add additional references, like Zhang et al. 2009a, here.

Tables and Figure: Table 1. Possible to include relative abundance of differenty fatty acids (parentheses after isotope values)?

The relative abundance of the different fatty acids is in Table S4. Adding it to Table 1 might make it quite packed and more difficult to keep the overview.

Figure 1. include C20:5 pufa as "calibrating" data for autotrophy

Due to the multiple sources with different isotopic values for this fatty acid, the changing contributions during the season the isotopic values of the C20:5 PUFA might not be suited as calibrating data for autotrophy. PUFA sources and potential reasons for its changing isotopic composition should be studied further before it can be used as "autotrophy" endmember. However, as discussed above, it would be nice to be able to include biomarker fatty acids for the different metabolisms to compare with the general fatty acids reflecting the entire community.

Fig 2: Do you have total cell counts? If so, maybe you can use this to get a rough estimate of the heterotroph cell abundance.

Unfortunately, as discussed above we don't have total cell counts of the heterotrophic bacteria.

Figure 4,5: add in bar highlighting bloom data, the most obvious trends in fractionation are the dips in C16:0 data, one corresponding to the bloom, and another dip in October. The fractionation goes as low as the bloom fractionation but is unrelated to photoautotrophy?

As mentioned already in our rebuttal to the comments of reviewer 1, we will add indications for the different season to the graphs.

---

## Author Comment (AC3) · 14 Jul 2016

Reviewer 3 (Alex Sessions) The purpose of this manuscript is to demonstrate a seasonal shift in the D values of marine fatty acids; such a pattern has not previously been reported. This manuscript is concise and well-written, and was a pleasure to read; the relative brevity belies a significant amount of effort that must have gone into the analysis of samples. The data all appear to be of high quality. The main conclusion is that the D value of average fatty acids gets more negative during the spring bloom; authors attribute this pattern to an increase in autotrophic contributions to particulate biomass during the spring, and greater heterotrophic input later in the summer. The attribution of this pattern to changing auto/heterotrophic inputs is very interesting, and

if it proves to be correct would be an exciting demonstration of the utility of lipid D values as ecological tracers. So I think the work is definitely significant. However, I don't think the case is as firmly settled as the current manuscript makes it appear. The main issue I have is that all of the fatty acids – including 20:5, which is a putative algal marker – show the same temporal pattern of depletion then enrichment over the year. Plus the use of 3um filters should preferentially collect algae rather than bacteria. So it seems entirely possible that the observed pattern is due (mainly or entirely) to changing fractionations by phytoplankton, rather than changing relative inputs from bacteria and algae. Perhaps the authors can think of some way to rule this hypothesis out; if not, I think it has to be presented as an alternative hypothesis that could explain the data.

First of all we want to thank the reviewer for the helpful comments. We see the problem with all fatty acids showing the same temporal pattern in depletion to enrichment and that therefore there is the possibility that the signal is due to changes in hydrogen isotopic fractionation in the phytoplankton and not so much the different input of photoautotrophs and heterotrophs. Unfortunately, we don't have any independent constrains on the relative abundance of heterotrophic bacteria versus autotrophic phytoplankton during our sampling period. However, as pointed out earlier Brandsma et al. 2012 studied this site earlier and reported algae, cyanobacteria and bacteria counts. They showed that in general the bacteria count was ten times the count of the algae/cyanobacteria and in general there were only changes in the number of algae and cyanobacteria cells with the two blooms and after the blooms the cell count dropped. For bacteria cells, the cell count was lowest during the spring bloom, increased directly after the bloom and stayed rather stable thereafter. Considering that we don't expect any drastic changes in this seasonal succession, we assume to have a similar situation during our sampling period. We are therefore quite confident that our signal is indeed due to changes in the relative input of photoautotrophs and heterotrophs. However we will add hypothesis that our isotopic signal due to changing fractionations by phytoplankton to the revised manuscript. As mentioned in the rebuttal to reviewer 1 we are aware that there are

bacteria which will be smaller than the 0.7 $\mu$m and will therefore not be captured by the filters we used. Unfortunately, for lipid analysis the filters used have to be plastic free and at the time of sampling there were no suitable filters available with a smaller pore size. The clogging of the 3 $\mu$m and the rather low fatty acid yield of the 0.7 $\mu$m filters lead to the assumption that we captured the majority of the biomass in the water samples. Pitcher et al. 2011 (Limnology and Oceanography) also showed in their work a good signal for the small Thaumarchaeota in 3 $\mu$m filters. Of course we cannot rule out that we still have a discrimination towards bigger cells and that a portion of the smaller cells might not have been captured and therefore analysed. Unfortunately, at the time of sampling there was no methodological possibility to change those biases.

Other minor comments: L56-57; its the hydrogen supplied by NADPH (not the whole molecule) that matters, and its the reduction of NADP+ to NADPH that matters, not its biosynthesis. I'm sure the authors know this, just be precise in the wording.

Thanks for the comment and we will change the wording in order to be more precise

L91: 'comprised of' rather than 'formed by'

This will be changed in the manuscript.

L137-138: I agree that the 3um filters can trap some bacteria as they get loaded, but still your sample will be strongly biased towards algae rather than bacteria. You might want to point this out, because the seasonal signal you observed could be even larger in a sample that more effectively traps bacterial biomass. Collecting bigger samples on smaller pore-size filters would also allow to test your assumption that bacterial lipids are D-enriched.

The North Sea water samples were first filtered through a 3 $\mu$m and then through a 0.7 $\mu$m filter. For three random samples both filter sizes were extracted and the fatty acid fractions analysed on GC respectively GC/irMS. Unfortunately, for all three samples the 0.7 $\mu$m filters did not yield enough material for isotope analysis. Therefore, we

assumed that the majority of the biomass did indeed end up in the 3 $\mu$m filters. We agree that there is a chance that the fatty acid results might be biased towards algae rather than bacteria and this will be noted in the revised manuscript.

L147. Boiling point of methanol is around 65C, so how did you reflux something at 140C in methanol? Would have to be under pressure...?

The was the temperature set on the heater for refluxing. We therefore left the temperature out, and simply say that the mixture was refluxed.

L183: I believe the internal standard (squalane) is allowing you to test accuracy, rather than precision. Precision is simply the reproducibility of unknown values. We changed that in the text.

L261-275. This is the one place where I thought the manuscript could be a bit shorter. The patterns of changing abundance are shown clearly in the figure, and not every detail needs to be recounted here in the text. Just point out the most important trends.

We thank the reviewer for the remark and we will shorten this part accordingly.

L309-311. I believe a similar trend (shortest-chain FA are most D-depleted) has been observed in many pure cultures and single (macro)organisms as well. Here it would be worth comparing the trend you see to some previously published results. Later, when you try to attribute C18:x to greater heterotrophic input, I would be cautious: the fact that it is D-enriched relative to 14:0 might be true even in algae, and would not necessarily require heterotrophic input. You could also point out that a similar pattern (increasing D with chain length, except for PUFA's) has been observed in other environmental samples, like Maggie's Yellowstone paper and Ashley Jones' Santa Barbara paper.

Thanks for the comment. It is indeed true that it has been observed in some, but not all, culture experiments that the shortest-chain FA was the most depleted. We will add a comment concerning this observation to the revised manuscript.

[Figure]

L347. I agree this is theoretically possible, but the timing is weird, because its in the summer (not fall). Why would plankton switch to more storage products in the middle of summer, when they have the most sunlight. Even if they are nutrient-limited they can still make sugars...

When the photoautotrophic organisms are active, but not growing, while still producing sugars (similar to stationary phase). When using these sugars via for example the pentose phosphate pathway for the reduction of NADP to NADPH, the newly synthesized fatty acids would become more enriched in D.

L354-355 This is where I'd be cautious; similar pattern seen in many individual organisms, I think.

We thank the reviewer for the comment and will mention in the manuscript that also the chain length might factor into the fact that the C14:0 is general the most depleted fatty acid.

L375. The statement that "C18:0 will be mainly derived from heterotrophic bacteria" is not supported by table S6, which just lists the relative abundance of fatty acids. If algal inputs are 3 orders of magnitude greater than bacteria, then the bulk C18:0 will still be mainly from the algae, even though bacteria have a higher relative abundance. Note that I am not saying it is not possible for C18 to be mainly bacterial, just that the data presented does not prove that point.

We appreciate the comment and agree with it. We will point out this possibility in the revised manuscript.

L377-378. I am currently working on a manuscript with Dave Valentine that compiles published fractionations for a variety of chemoautotrophs, and shows that they are not statistically significantly different from photoautotrophs. The first few organisms that were studied, that yielded such huge depletions, turn out to be anomalous. So I would probably just delete this sentence.

We appreciate the remark and look forward to read the manuscript when it is published. The sentence will be deleted.

L380: You are comparing epsilon values (from current study) to the D values from Sandra's earlier culture work. The epsilon values from her study for chemoautotrophs are -217 to -275‰ (numbers taken from the abstract).

We actually meant here the $\varepsilon$ values for chemoautotrophs summarized in figure 3 of Heinzelmann et al. 2015b here. As remarked with reviewer 2 we will add other references here.

L393. I think you mean Figure 1 here? Fig 5 does not show data for the summed fatty acids.

Thanks for the note. Indeed we mean Figure 1 and this will be corrected in the manuscript.